# Identification of Dysregulated microRNAs in Glioblastoma Stem-like Cells

**DOI:** 10.3390/brainsci13020350

**Published:** 2023-02-18

**Authors:** Lara Evers, Agnes Schäfer, Raffaella Pini, Kai Zhao, Susanne Stei, Christopher Nimsky, Jörg W. Bartsch

**Affiliations:** 1Department of Neurosurgery, Philipps-University Marburg, University Hospital Marburg (UKGM), Baldingerstrasse, 35043 Marburg, Germany; 2Center for Omics Sciences, IRCCS San Raffaele Scientific Institute, Via Olgettina 58, 20132 Milan, Italy; 3Marburg Center for Mind, Brain and Behavior (MCMBB), 35032 Marburg, Germany

**Keywords:** glioblastoma multiforme, glioblastoma stem-like cells, differentiation, microRNA, GFAP, miR-425-5p, miR-223-3p, let-7, miR-17-5p

## Abstract

Glioblastoma multiforme (GBM) is the most common malignant primary brain tumor in adults. Despite multimodal therapy, median survival is poor at 12–15 months. At the molecular level, radio-/chemoresistance and resulting tumor progression are attributed to a small fraction of tumor cells, termed glioblastoma stem-like cells (GSCs). These CD133-expressing, self-renewing cells display the properties of multi-lineage differentiation, resulting in the heterogenous composition of GBM. MicroRNAs (miRNAs) as regulators of gene expression at the post-transcriptional level can alter many pathways pivotal to cancer stem cell fate. This study explored changes in the miRNA expression profiles in patient-derived GSCs altered on differentiation into glial fiber acid protein (GFAP)-expressing, astrocytic tumor cells using a polymerase chain reaction (PCR) array. Initially, 22 miRNAs showed higher expression in GSCs and 9 miRNAs in differentiated cells. The two most downregulated miRNAs in differentiated GSCs were miR-17-5p and miR-425-5p, whilst the most upregulated miRNAs were miR-223-3p and let-7-5p. Among those, miR-425-5p showed the highest consistency in an upregulation in all three GSCs. By transfection of a 425-5p miRNA mimic, we demonstrated downregulation of the GFAP protein in differentiated patient-derived GBM cells, providing potential evidence for direct regulation of miRNAs in the GSC/GBM cell transition.

## 1. Introduction

Glioblastoma multiforme, the most common malignant primary brain tumor in adults, is characterized by an aggressive and invasive growth pattern, rapid development of radio-/chemoresistance, and genetic heterogeneity [1]. The current therapeutic standard of care consists of maximum safe surgical resection, radiation, and temozolomide (TMZ) chemotherapy [2]. However, the median survival remains low at 12–15 months as tumor recurrence occurs rapidly [3].

Glioblastoma stem-like cells (GSCs) are currently viewed as modulators of the tumor microenvironment as well as the origin of radio-/chemoresistance thereby resulting in tumor progression [4]. Due to this small, but pluripotent self-renewing subpopulation of GBM tumor cells that typically reside in perivascular niches apart from the bulk tumor mass, GSCs cannot be sufficiently targeted by surgical resection [5]. As a result of GBM heterogeneity due to different GBM phenotypes, such as the classical, proneural, and mesenchymal types, patient-derived cell-cultured GSC lines might display diverging characteristics [6]. At the molecular level, GSCs express a unique pattern of stemness markers such as the transmembrane glycoproteins CD44 and CD133 or the transcription factor Sex-determining region Y-box2 (SOX2) [4,7]. In particular, CD133, an established marker for neural progenitor cells and cancer stem-like cells, organizes the cell membrane topology [8]. Contrary to the bulk mass of astrocyte tumor cells, GSCs barely express the intermediate filament GFAP [9]. Therefore, GFAP is utilized as a marker for primary differentiated, astrocytic tumor cells in GBM [10]. 

Not only does the expression pattern of proteins change on GSC differentiation but recent studies also suggest that GSCs display a unique miRNA expression pattern [11]. MicroRNAs (miRNAs) as small noncoding RNA molecules regulate gene expression at the posttranscriptional level by binding to and thereby targeting their corresponding mRNAs [12]. As miRNAs can mediate many critical pathways to cancer progression such as proliferation, apoptosis, and angiogenesis, they can act as both tumor-suppressors (tumor-suppressor miRNAs) and oncogenes (onco-miRNAs) [13]. Clinically, miRNAs are receiving rising attention as their function as novel diagnostic and prognostic biomarkers, as well as future therapeutic agents, is discussed and given that they can affect multiple target genes involved in pathological processes [11]. 

For this reason, dysregulated miRNAs are intensely studied in GBM. However, the expression profile as well as the function of specific miRNAs in GSCs have not yet been adequately elucidated. 

This study investigated changes in the miRNA expression profile in patient-derived, well-characterized, cultured, sphere-forming GSCs and their differentiated status as adherent GBM cells by utilizing a miRNA PCR array. As a result, a total of 31 dysregulated miRNAs were identified. Through a literature review and target prediction analyses, we closely investigated the most dysregulated miRNAs.

## 2. Materials and Methods

### 2.1. Cell Culture

After approval from the local ethics committee (Philipps University Marburg, medical faculty, file number 185/11), patient-derived GSCs as well as primary GBM cell lines were obtained during surgical resection. Each patient gave written informed consent before surgical resection. Isolation, preparation, and molecular characteristics of GSCs and primary GBM cell lines from resected tumor tissues were described previously [14,15]. GSC lines 2017/151, 2017/74, and 2016/240 were cultivated in non-cell-culture-treated Petri dishes. As a medium, DMEM/F12 (DMEM-12-A, Capricorn Scientific, Ebsdorfergrund, Germany), supplemented with 2% B27 (17504044, Thermo Fisher Scientific, Waltham, MA, USA), 1% amphotericin (15290026, Thermo Fisher Scientific, Waltham, MA, USA), 0.5% HEPES (H0887, Sigma-Aldrich, Taufkirchen, Germany), and 0.1% gentamycin (A2712, Biochrom, Berlin, Germany), was utilized. In addition, epidermal growth factor (EGF, 100-15, Peprotech, Hamburg, Germany) and basic fibroblast growth factor (bFGF, 100-18b, Peprotech, Hamburg, Germany) were both supplemented at a final concentration of 0.02 ng/μL. Primary differentiated GBM cell lines GBM100 and GBM42 were cultivated in phenol red-free DMEM (DMEM-HXRXA, Capricorn Scientific, Ebsdorfergrund, Germany) supplemented with 10% fetal calf serum (FCS, S0615, Sigma, Taufkirchen, Germany), 1% penicillin/streptomycin (2321115, Gibco, Carlsbad, CA, USA), 1 mM sodium pyruvate (NPY-B, Capricorn Scientific, Ebsdorfergrund, Germany), 1% L-glutamine (25030-024, Gibco, Carlsbad, CA, USA), and 1% non-essential amino acids (11140050, Gibco, Carlsbad, CA, USA). All cell lines were cultivated in a humidified atmosphere at 37 °C and 5% CO_2_. 

### 2.2. GSC Differentiation

To differentiate GSCs, cells of 2016/240, 2017/151, and 2017/74 were seeded in 6-well plates at a density of 750,000 cells in 2 mL. To initiate GSC differentiation, 10% FCS (S0615, Sigma-Aldrich, Taufkirchen, Germany) was supplemented in DMEM/F12. In addition, bFGF and EGF were withdrawn. After seven days of incubation, light-microscopy images were taken. Then, cells were harvested for further analyses.

### 2.3. miR-425-5p Mimic Transfection

For transient overexpression of miR-425-5p, primary GBM100 and GBM42 cell lines were transfected with 0.01 µM hsa-miR-425-5p miRCURY LNA miRNA (GeneGlobe ID: YM00471725-ADA, catalog no.: 339173, Qiagen, Hilden, Germany). In detail, cells were seeded in a 6-well format at a density of 300,000 cells in 2 mL. After 24 h of incubation and attaching, the transfection was performed with Lipofectamine 2000 Reagent (11668-030, Thermo Fisher Scientific, Waltham, MA, USA) according to the manufacturer’s instructions. Meanwhile, 0.01 µM Allstar Negative Control siRNA (1027280, Qiagen, Hilden, Germany) was transfected as a control. The transfection was repeated after 24 h. Cells were harvested 48 h after the second transfection and further analyzed via qPCR and Western blot.

### 2.4. RNA and miRNA Isolation

Total RNA with an enriched fraction of miRNAs from cellular pellets was isolated using the miRNeasy Tissue/Cells Advanced Mini Kit (217684, Qiagen, Hilden, Germany) according to the manufacturer’s instructions. 

### 2.5. RNA Reverse Transcription (RT) and Quantitive Real-Time Polymerase Chain Reaction (qPCR)

To quantify gene expression on an mRNA level, total RNA was reverse transcribed using the RNA to cDNA EcoDryTM Premix (639548, TaKaRa, Saint-Germain-en-Laye, France) according to the manufacturer’s instructions. Quantitative real-time PCR was performed with a total reaction volume of 20 µL/well, consisting of 10 µL SYBR Green/Rox Master Mix (PPLUS-R-10 ML, Primer Design, Eastleigh, UK), 2 μL *GFAP*/*CD133* primers (244900, Qiagen, Hilden, Germany), 6 μL nuclease-free water, and 2 μL cDNA. Expression of the ribosomal gene *RPLP0*/*XS13* (fw: 5′-TGG GCA AGA ACA CCA TGA TG-3′; rev: 5′-AGT TTC TCC AGA GCT GGG TTG T-3′) was used as a housekeeping gene for normalization [16]. PCR experiments were performed on the Applied Biosystems StepOnePlus Real-Time PCR System (Thermo Fisher Scientific, Waltham, MA, USA). Relative gene expression was calculated utilizing the 2^−∆∆Ct^ method.

### 2.6. miRNA Reverse Transcription and miRNA PCR Array

First, pooled samples for GSCs and differentiated GBM cells consisting of 8.3 ng total RNA with an enriched miRNA fraction from each of the three cell lines (2017/151, 2016/240, and 2017/74) were generated. As a next step, reverse transcription of the pooled samples was performed utilizing a miScript II RT Kit (218161, Qiagen, Hilden, Germany). Then, following the manufacturer’s instructions, a pathway-focused miRNA PCR array (331221 miScript, Qiagen, Hilden, Germany) was conducted utilizing the miScript SYBR Green PCR Kit (218073, Qiagen, Hilden, Germany). The miRNA PCR arrays were performed on the Applied Biosystems StepOnePlus Real-Time PCR System (Thermo Fisher Scientific, Waltham, MA, USA). Data analysis and scatter plot generation were performed using the corresponding online data analysis tool provided by Qiagen (miScript miRNA PCR Data Analysis, Qiagen, https://dataanalysis.qiagen.com/mirna/arrayanalysis.php?target=upload, accessed on 1 February 2023). Relative gene expression was calculated utilizing the 2^−∆∆Ct^ method. RNU6 was used for internal normalization. Results are presented in heatmaps, which were generated using the GraphPad PRISM 9 software, version 9.1 (Insight Partners, New York, NY, USA).

### 2.7. miRNA Reverse Transcription and qPCR

For verification of the miRNA PCR array results and further functional experiments on miR-425-5p, isolated RNA samples with an enriched fraction of miRNAs were reverse transcribed utilizing the miRCURY LNA RT Kit (339340, Qiagen, Hilden, Germany), according to the manufacturer’s instructions. Quantitative real-time PCR was performed utilizing the miRCURY LNA SYBR^®^ Green PCR Kit (339345, Qiagen, Hilden, Germany), according to the manufacturer’s instructions. As miRNA primers, hsa-miR-17-5p miRCURY LNA miRNA PCR Assay (YP02119304, 339306, Qiagen, Hilden, Germany), hsa-miR-425-5p miRCURY LNA miRNA PCR Assay (YP00204337, 339306, Qiagen, Hilden, Germany), hsa-miR-223-3p 5p miRCURY LNA miRNA PCR Assay (YP00205986, 339306, Qiagen, Hilden, Germany), and hsa-miR-7a-5p miRCURY LNA miRNA PCR Assay (YP00205727, 339306, Qiagen, Hilden, Germany) were used. For normalization, miR-24-5p (YP00203954, 339306, Qiagen, Hilden, Germany) and UniSp6 (YP00203954, 339306, Qiagen, Hilden, Germany) were used. Relative gene expression was calculated utilizing the 2^−∆∆Ct^ method.

### 2.8. Protein Extraction and Western Blot Analysis

For 30 min, whole cell lysates were incubated in RIPA buffer (50 mM HEPES pH 7.4; 150 mM NaCl; 1% (*v*/*v*) NP-40; 0.5% (*w*/*v*) natriumdeoxycholate; 0.1% (*w*/*v*) SDS; 10 mM phenantrolin; 10 mM EDTA; PierceTM Protease Inhibitor Mini Tablets, EDTA-free, Thermo Fisher Scientific; PierceTM Phosphatase Inhibitor Mini Tablets, Thermo Fisher Scientific). Then, protein samples were prepared in 5× Laemmli buffer (60 mM Tris HCl, pH: 6.8; 2% (*w*/*v*) SDS; 10% (*w*/*v*) glycerol; 5% (*v*/*v*) ß-mercaptoethanol; 0.01% (*w*/*v*) bromophenol blue) and 10× NuPAGETM sample reducing reagent (Thermo Fisher Scientific, Waltham, MA, USA). To separate proteins, samples were denatured at 95 °C for 5 min, then 12.5% SDS polyacrylamide gel was utilized for separation. Separated proteins were transferred onto nitrocellulose membranes (A29591442, GE Healthcare Life Science, Munich, Germany) followed by blocking in 5% (*w*/*v*) milk powder (MP) in TBST (50 mM Tris, pH 7.5; 150 mM NaCl; 0.1% (*w*/*v*) Tween-20), and then incubated for 1 h. The following primary antibodies were utilized: anti-PTEN (1:1000 in 5% MP in TBST, 9559T, Cell Signaling, Leiden, NL, USA), anti-GFAP (1:1000 in 5% bovine serum albumin in TBST, M0761, Dako GmbH, Jena, Germany), anti-SOX2 (1:2000 in 5% MP in TBST, ab97959, Abcam, Berlin, Germany), and anti-GAPDH (glyceraldehyde 3-phosphate dehydrogenase, 1:10,000 in 5% MP in TBST, 181602, Abcam, UK). After overnight incubation with primary antibodies at 4 °C, nitrocellulose membranes were washed three times with TBST. Then, membranes were incubated with horseradish peroxidase-conjugated antibodies (ab2116, Abcam, 1:5000) for 1 h. Membranes were washed again with TBST. By the addition of Western Bright Sirius substrate (K-12043-D10, Advansta, San Jose, CA, USA), chemiluminescence was detected using the ChemiDoc MP Imaging System (Bio-rad Laboratories GmbH, Feldkirchen, Germany). Western blot quantification was realized by using the Image J software version 1.53t (NIH, Bethesda, MD, USA). 

### 2.9. Kyoto Encyclopedia of Genes and Genomes (KEGG) Analyses

To functionally characterize the most dysregulated miRNAs in patient-derived GSCs and their differentiated status, KEGG enrichment analysis was performed using the DIANAmiRPath v3.0 web app, an online software suite dedicated to the evaluation of the regulatory role of miRNAs and the identification of controlled pathways [17]. The barplot and the chord diagram were built in the R environment (v. 4.1.3) with ggplot2, and circlize R packages [18,19].

### 2.10. Statistical Analysis

Results from multiple replicates are presented as the mean ± standard deviation (SD). The miRNA PCR array was conducted once. Paired Student’s *t*-tests were applied for statistical comparison between the two groups. Results were considered as not significant (ns) (*p* > 0.05), significant (*) (*p* < 0.05), highly significant (**) (*p* < 0.01), or very highly significant (***) (*p* < 0.001) / (****) (*p* < 0.0001). Statistical analysis was performed utilizing GraphPad PRISM 9, version 9.1 (Insight Partners, New York, NY, USA).

## 3. Results

### 3.1. Differentiation of GSCs

GSC lines 2017/151, 2016/240, and 2017/74 were derived from resected tumor tissues of three patients with primary, isocitrate-dehydrogenase (IDH) wildtype GBM. Information regarding molecular-pathological features as well as clinical information is presented in Table 1. In cell culture, GSCs formed typical non-adherent neurospheres (Figure 1a, left). On differentiation, cells acquired morphological features similar to those of glial cells. For instance, differentiated cells grew in monolayers attached to the bottom of six-well plates and developed long, star-shaped cellular protrusions (Figure 1a, right). As previously demonstrated, a side population analysis was conducted. Here, a population of cells with a higher efflux, hence a lower intracellular concentration of Hoechst dye, was identified. Inhibition of ABC transporters with verapamil and concomitant blockage of efflux confirmed the specificity of the side population as an efflux was no longer detectable [14]. At the molecular level, GSCs expressed high levels of the stem cell marker *CD133*. In contrast, differentiation resulted in a significant decrease in *CD133* expression on the mRNA level in all three GSC lines (Figure 1b, left). On the mRNA level, additional stem cell markers such as CD44, Sox2, and Nestin were tested with similar, but less consistent trends for Sox2 and Nestin, while CD44 was induced in differentiated GSCs (Appendix A), similar to the significant increase observed for *GFAP* mRNA expression (Figure 1b, right).

In addition, Western blot analysis demonstrated that the stem cell marker SOX2 is more greatly expressed in GSCs compared to differentiated cells, while GFAP protein expression increases on differentiation (Figure 1c,d). To summarize, these data suggest that all three GSC lines were successfully differentiated into adherent, growing, astrocytic tumor cells, although to different extents. 

### 3.2. Identification of Dysregulated miRNAs in GSCs and Differentiated Cells

As a screening method to identify changes in miRNA expression induced by GSC differentiation, a pathway-focused miRNA PCR array was conducted. This was realized by generating pooled samples of either GSCs or differentiated cells, containing an equal concentration of miRNAs from each of the three GSC lines and differentiated cell lines, respectively. Differences in miRNA expression based on fold regulation are presented for 84 tested miRNAs by a heatmap (Figure 2a). While green signals represent upregulation in GSCs, the red color indicates the downregulation of respective miRNAs in GSCs and consequently higher expression in differentiated cells. All miRNAs exhibiting fold regulation values > 2 or <−2 were interpreted to be dysregulated by the Qiagen analysis tool. A scatter plot analysis revealed that from a total of 84 tested miRNAs, 22 miRNAs were more greatly expressed in GSCs compared to differentiated cells. In contrast, nine miRNAs displayed lower expression in GSCs and were consequently more greatly expressed in differentiated cells (Figure 2b). A detailed analysis of these dysregulated miRNAs depicted by a heatmap revealed that 10 out of 31 miRNAs were particularly strongly dysregulated (Figure 2c and Table 2). Notably, miR-425-5p, miR-17-5p, miR-424-5p, miR-195-5p, and miR-30c-5p were highly expressed in GSCs. Meanwhile, miR-223-3p and four members of the let-7 miRNA family displayed higher expression in differentiated cells. A thorough literature research was conducted on these 10 miRNAs. Here, we focused on the general role of each miRNA in GBM and the current status of research concerning GSCs (Table 2). 

Taken together, the PCR array identified a changed miRNA expression profile in GSCs and differentiated cells and revealed several candidate miRNAs for further investigation into GSC/GBM cell transition. 

### 3.3. miR-425-5p Is Downregulated in Differentiated GSCs

From all dysregulated miRNAs, four miRNAs, miR-425-5p, miR-17-5p, let-7a-5p, and miR-223-3p, turned out to be functional candidates in GSCs for tumor cell differentiation. To validate the miRNA expression levels found in our PCR array screening using pooled miRNAs from all three GSCs (Figure 3a), we determined miRNA expression levels in all three GSC lines separately (Figure 3b and Appendix A). Most consistently, miR-425-5p was significantly overexpressed in all three GSC lines compared to differentiated, astrocytic tumor cells (Figure 3b). miR-17-5p proved to be significantly overexpressed in GSCs (Appendix A). Meanwhile, let-7a-5p and miR223-3p downregulation in GSCs compared with differentiated cells could not be verified in the three GSC lines (Appendix A).

### 3.4. Transfection of miRNA Mimic Affects Protein Levels of GFAP in Patient-Derived GBM Cells

As shown in Table 2, miR-425-5p was identified to potentially target the GFAP gene, which is expressed as a marker of differentiated GBM cells, so that downregulation of miR-425-5p in GSCs could increase GFAP levels in the course of differentiation into GBM cells. To explore this, we transfected two patient-derived GBM cell strains (GBM100 and GBM42) with a miR-425-5p mimic miRNA (Figure 3c). About 48 h after transfection, cells were lysed and analyzed for protein levels of GFAP and the known miR-425-5p target PTEN. In GBM100, both GFAP and PTEN protein levels were reduced after mimic transfection (Figure 3d,e), whereas the results for GBM42 were inconsistent (Figure 3f,g). Those first results suggested that GFAP is a target gene for miR-425-5p and could potentially regulate the differentiation state of GBM cells.

### 3.5. miRNA Profiling in GSC Maintenance and Differentiation

The 10 most regulated miRNAs were analyzed for their biological functions by KEGG enrichment analysis (Figure 4a). The most significant pathways were “Signaling pathways regulating pluripotency of stem cells”, “Pathways in cancer”, and “PI3K-Akt signaling”, a pathway of high importance in GBM. In Figure 4b, the miRNA target relationship to pathways regulating the pluripotency of stem cells is shown in the form of a chord plot. All 10 miRNAs are connected to their mRNA targets, thereby showing their contribution to GSC maintenance or differentiation.

## 4. Discussion

Due to tumor heterogeneity and limited therapeutic options, GBM remains an incurable disease with a devasting prognosis. As modulators of the tumor microenvironment and radio-/chemoresistance, GSCs are considered putative future therapeutic targets [6]. Even though GSCs play a key role in tumor cell invasion, recurrence, and angiogenesis, many underlying signaling pathways remain elusive [41]. MicroRNAs (miRNAs) can act as central regulatory molecules of GBM hallmarks such as invasion or immune evasion [42]. Therefore, miRNAs are discussed as future therapeutic targets and diagnostic biomarkers [43]. Our study detected differences in the miRNA expression profile of GSCs and differentiated tumor cells by PCR array screening. Cultured patient-derived GSCs were stimulated to differentiate into astrocytic tumor cells. Although this in vitro differentiation model is commonly used in GSC research, in vivo GSC differentiation is a complex process as the tumor microenvironment is shaped by a variety of different cell types such as macrophages, microglia, and mesenchymal cells [44].

Based on our results from miRNA PCR array screening, a literature review of mRNA targets, and validating experiments in each GSC line, we present four suitable miRNA candidates, miR425-5p, miR-17-5p, miR-223-3p, and let-7a-5p, which might be directly and indirectly involved in the regulation of GBM cell differentiation.

Firstly, miR-17-5p, the most dysregulated miRNA in the conducted array, is a known onco-miRNA in GBM [20]. Consistent with other studies, miR-17-5p was highly expressed in GSCs as it stimulates GSC proliferation [21,22,23]. As physically adjacent miRNA genes are often transcribed at the same time, they are summarized as a cluster. miR-17-5p is often analyzed as a part of the miR-17-92 cluster [45]. Notably, four of six miRNAs of the miR-17-92 cluster were consistently upregulated in our conducted PCR array. In GBM, the miR-17-92 cluster is highly expressed and correlated with a poor prognosis [20].

Secondly, miR-425-5p was upregulated in GSCs concordant with a study by La Rocha et al. [25]. MiR-425-5p is overexpressed in GBM tissue specimens in comparison to normal brain control specimens and acts as an onco-miRNA [25]. Both miR17-5p and miR-425-5p are known to target phosphatase and tensin homolog (PTEN) mRNA [23,26]. PTEN, a key tumor suppressor, is commonly mutated in GBM carcinogenesis [46]. As a predicted mRNA target based on miRPathDB v2.0, expression levels of GFAP could be regulated directly by miR-425-5p and miR-17-5p, suggesting that these miRNAs control astrocytic cell differentiation [24]. As an experimental proof, mimic miR-425-5p when transfected into differentiated patient-derived GBM cells, can regulate GFAP expression. Since these cells are also able to de-differentiate into GSCs, we interpret our findings to reveal that miR-425-5p is able to contribute, among other critical proteins, to GBM cell differentiation. Moreover, as chord analysis suggests, miR-425-5p is potentially involved in gene regulation of IGF1 (insulin growth factor 1), gp 130 (IL6ST), MEIS1 (a homeobox gene), SMAD5 (TGF-ß pathway), and PCGF5 (a polycomb transcription factor). All of them could be important mediators of growth signals and cell fate determination in GBM cells.

Furthermore, PCR array analysis identified nine miRNAs that were upregulated in differentiated cells. Notably, six of these nine miRNAs are members of the let-7 miRNA family. Published data revealed that the let-7 family acts as a tumor suppressor in GBM [11,36]. Overexpression of let-7 miRNAs leads to the inhibition of tumor cell migration and promotes apoptosis [36]. According to our PCR array, Degrauwe et al. demonstrated that the let-7 family is scarcely expressed in GSCs [38]. As an interesting mRNA target, Kirsten rat sarcoma virus oncogene homolog (K-Ras), an oncogene and activator of its downstream targets in the mitogen-activated protein kinase (MAPK) pathway, was identified [38]. According to miRPathDB v.2.0 target gene prediction, the stem cell marker Musashi-2 can be directly regulated by the let-7 family and miR-223-3p so that high expression of these miRNAs could suppress the GSC phenotype [24,47]. Additionally, miR-223-3p was overexpressed in differentiated cells. In GBM, miR-223-3p functions as a tumor-suppressor miRNA; its overexpression enhances radio-/chemosensitivity in cell culture models as miR-223-3p targets ataxia telangiectasia mutated (ATM) [40]. ATM initiates repair mechanisms after radio-/chemotherapy-induced DNA damage and thereby contributes to radio-/chemoresistance [48]. Since the role of miR-223-3p is currently uninvestigated in GSCs, future detailed analysis of miR-223-3p and ATM in GSCs is justified.

## 5. Conclusions

Our study detected a changed miRNA expression profile on GSC differentiation in a well-defined in vitro setup. Through a miRNA PCR array, 31 dysregulated miRNAs were identified. About 10 highly regulated miRNAs, including miR-425-5p, miR-17-5p, miR-223-3p, and the let-7 miRNA family, are promising miRNA candidates for further investigations aiming to manipulate the differentiation status of GSCs.

## Figures and Tables

**Figure 1 brainsci-13-00350-f001:**
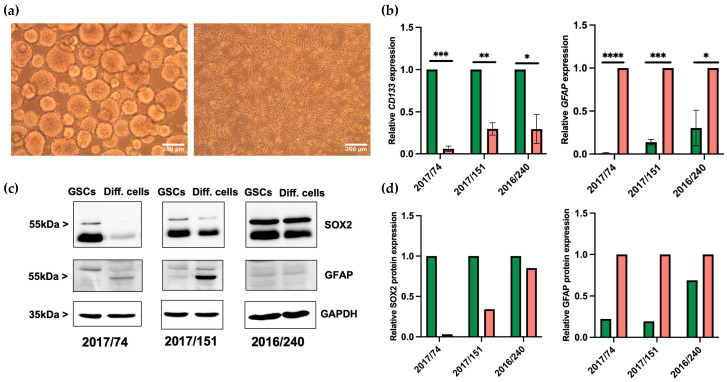
Differentiation of GSCs in astrocytic tumor cells. (**a**) Light microscopy images of 2017/151 spheroid GSCs (left) and adherent differentiated cells (Diff. cells, right). (**b**) Expression of *CD133* and *GFAP* on an mRNA level by RT-qPCR in GSCs (green bars) and corresponding differentiated, astrocytic cells (red bars). Results are given as mean ± SD of three independent experiments. A paired Student’s *t*-test was applied to determine significance: * *p* < 0.05, ** *p* < 0.01, *** *p* < 0.001, **** *p* < 0.0001. (**c**) Western Blot of GSCs and Diff. cells showing SOX2 and GFAP expression, where GAPDH was used for internal normalization. (**d**) Western Blot quantification of stem cell marker SOX2 and differentiation marker GFAP in three GSCs lines (green bars) and differentiated cells (red bars).

**Figure 2 brainsci-13-00350-f002:**
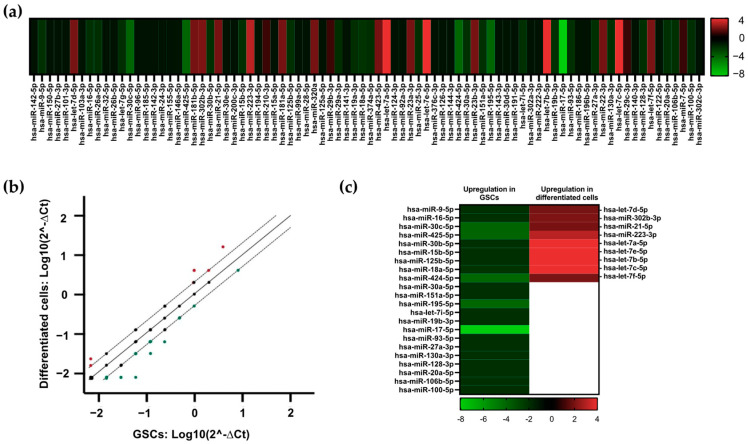
Differentially expressed miRNAs in GSCs and differentiated GBM cells. (**a**) Heatmap of differentially expressed miRNAs in pooled GSCs and pooled differentiated cells generated by fold expression values. Fold regulation values > 1 indicate lower miRNA expression in GSCs and overexpression in differentiated cells (red). Fold regulation values < 0 indicate higher expression in GSCs in comparison with differentiated cells (green). (**b**) Scatter plot analysis (log10 of 2^−delta Ct) of 84 miRNAs tested by the miRNA PCR array. Dotted lines equal log10 of fold regulation of 2 and −2. Green dots indicate upregulation in GSCs, black dots indicate no dysregulation, and red dots indicate overexpression in differentiated cells. (**c**) Heatmap of all tested miRNAs exhibiting fold regulation > 2 or < −2. Fold regulation values > 2 indicate miRNA overexpression in differentiated cells compared to GSCs (red). Fold regulation < −2 indicates miRNA overexpression in GSCs (green).

**Figure 3 brainsci-13-00350-f003:**
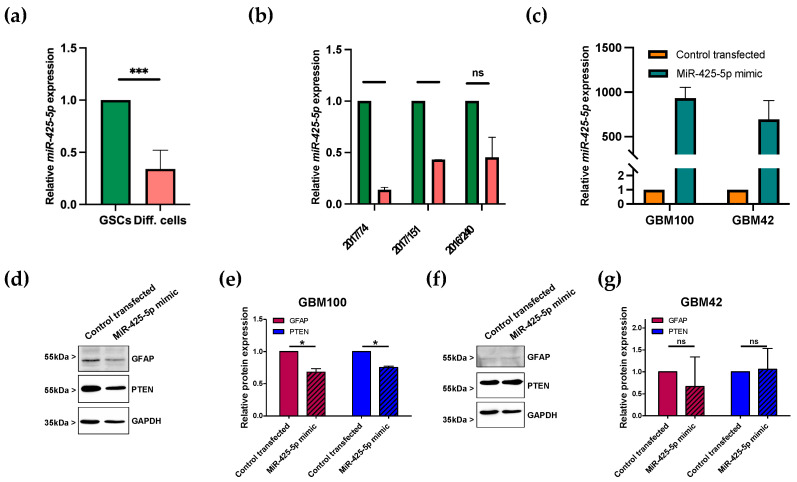
Expression of miR-425-5p in GSCs and effects on GFAP and PTEN protein levels in GBM cells. (**a**) Expression of miR-425-5p in GSCs (green) and their differentiated state (red). RT-qPCR results for differentiated 2017/74, 2017/151, and 2016/240 were normalized to undifferentiated controls. (**b**) Detailed depiction of miR-425-5p expression in each GSC line (green) and its corresponding differentiated state (red). Results are shown as mean values ± SD of two independent experiments. (**c**) miR-425-5p mimic transfection of primary GBM cell lines GBM100 and GBM42. (**d**) Representative Western blot demonstrating GFAP and PTEN expression after miR-425-5p mimic transfection in GBM100 cells. (**e**) Quantification of GFAP and PTEN protein expression in transfected GBM100 cells. (**f**) Representative Western blot demonstrating GFAP and PTEN expression after miR-425-5p mimic transfection in GBM42 cells. (**g**) Quantification of GFAP and PTEN protein expression in transfected GBM42 cells. Two independent experiments were conducted. Results are presented as mean values ± SD. A paired Student’s *t*-test was applied to determine significance: ns *p* > 0.05, * *p* < 0.05, *** *p* < 0.001.

**Figure 4 brainsci-13-00350-f004:**
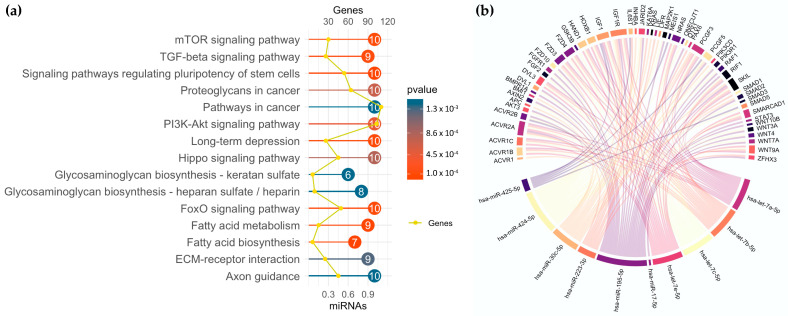
Bioinformatic analysis of the 10 most dysregulated miRNAs. (**a**) KEGG enrichment analysis of the 10 most dysregulated miRNAs. The x-axis reports the number of target genes and the fraction of miRNAs in the starting list involved in the pathway. The number within the dots represents the number of miRNAs. (**b**) The miRNA–target relationship in the KEGG_hsa04550 signaling pathway, “Signaling pathways regulating pluripotency of stem cells”. Each link represents a miRNA–target interaction.

**Table 1 brainsci-13-00350-t001:** Clinical information and histopathological characteristics of patient-derived GSC lines. All three patients suffered from primary, isocitrate dehydrogenase (IDH) wildtype glioblastoma. Here, clinical parameters including age at diagnosis, sex, survival in days, and tumor localization are presented. Furthermore, histopathological data such as methylation status of the O6-methylguanine-DNA-methyltransferase (MGMT), p53 accumulation, and Ki67 labeling index (Ki67-Li) are presented.

GSC Line	Age at Diagnosis (in Years)	Sex	Survival in Days	Localization	MGMT Promotor Methylation Status	Ki67-Li	p53 Accumulation
2016/240	48	female	641	right frontal lobe	methylated	up to 10%	moderately accumulated
2017/151	66	male	126	right temporal lobe and right insula	methylated	up to 20%	accumulated
2017/74	61	male	398	right temporal lobe	not methylated	up to 50%	moderately accumulated

**Table 2 brainsci-13-00350-t002:** Literature review results on highly dysregulated miRNAs identified in the PCR array. The table includes PCR array results indicated by fold regulation and published data on highly dysregulated miRNAs. We focused on the general role of each miRNA in GBM and their potential roles in GSCs. Plus, reported functionally relevant mRNA targets and predicted target genes directly involved in a GSC or differentiated state, respectively, are listed. For miRNA target prediction, the online software miRPathDB v2.0 was used.

miRNA	Upregulated in	Fold Regulation	Role in GBM	Role in GSCs	mRNA Target
miR-17-5p	GSCs	−8.05	Onco-miRNA [20] Highly expressed in GBM, correlated with poor prognosis [20]	Highly expressed in GSCs [21,22,23] Increases GSC proliferation [21]	PTEN [23] GFAP (predicted) [24]
miR-425-5p	GSCs	−4.00	Onco-miRNA [25] Associated with poor prognosis [25]	Highly expressed in GSCs [25] Promotes neurosphere formation and GSC survival [25]	PTEN [26] GFAP (predicted) [24]
miR-30c-5p	GSCs	−4.00	Conflicting data Promotes chemoresistance [27] Inhibition of proliferation, migration, and invasion [28] Downregulation in GBM tissue [28]	Unexplored	SOX-9 [28]
miR-424-5p	GSCs	−4.04	Conflicting data Effects on migration and proliferation, induction of apoptosis [29,30,31] Inhibition of epithelial-to-mesenchymal transition (EMT) and tumor growth [31] Enhances chemoresistance [30]	Unexplored	Akt-1, RAF1 [29] GFAP (predicted) [24]
miR-195-5p	GSCs	−4.05	Conflicting data Affects response to TMZ [32,33] Inhibits proliferation [34] Upregulated in recurrent GBM samples [35]	Unexplored	Cyclin E1 [32] Cyclin D1 [34] GFAP (predicted) [24]
let-7a-5p	Differentiated cells	4.01	Tumor-suppressor miRNA family [36,37] Inhibition of tumor cells’ migration, proliferation, and invasion [36,37] Promotes cell cycle arrest and apoptosis [36]	Low expression in GSCs [38] Inhibition of neurosphere growth [37]	K-Ras [36] Musashi-2 (predicted) [24]
let-7e-5p		3.97			MMP9 [39] Musashi-2 (predicted) [24]
let-7b-5p		3.97			E2F2 [37] Musashi-2, Musashi-1 (predicted) [24]
let-7c-5p		4.00			Musashi-2 (predicted) [24]
miR-233-3p	Differentiated cells	3.09	Tumor-suppressor miRNA [40] Enhances radiation sensitivity of GBM cells [40]	Unexplored	ATM [40] Musashi-2 (predicted) [24]

## Data Availability

Data presented in this study are available on reasonable request.

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
