# Peer review of "Identification of Dysregulated microRNAs in Glioblastoma Stem-like Cells"

_brainsci, 2023, doi:10.3390/brainsci13020350_

Round 1
Reviewer 1 Report
GENERAL COMMENTS:
- This manuscript identifies simple, but relevant information about GSCs Biology when comparing with bulk of tumour non-stem cancer cells.
- However, the authors do not go further in their findings, there are no functional experiments to validate the results of miRNA PCR array nor other validations of the findings described when comparing GSCs versus differentiated bulk of tumour cells. This makes the manuscript a quite descriptive work without confirmation of theses statements.
- Globally there is a lack of experimental work, most conclusions and discussion are based on bibliography research.
Specific comments:
1. In the introduction (line 39) the authors describe/define GSCs as cells that express a unique marker (CD133). This is a poor definition of GSCs; a better and wider contextualization of these cells is needed, in terms of a combination of different stem cells markers, heterogeneity within the stem cell compartment and/or the natural biology for the transition from Proneural to Mesenchymal upon GBM progression.
2. Table 1 itself and also the table 1 legend are allocated in the Materials and Methods section, the should move to Results.
3. Table 2 legend also should be moved to the bottom of Table 2.
Author Response
Dear Reviewer,
thanks for your productive comments on our manuscript. In our revised version, we have addressed the general and specific comments as follows:
GENERAL COMMENTS:
- However, the authors do not go further in their findings, there are no functional experiments to validate the results of miRNA PCR array nor other validations of the findings described when comparing GSCs versus differentiated bulk of tumour cells. This makes the manuscript a quite descriptive work without confirmation of theses statements.
- Globally there is a lack of experimental work, most conclusions and discussion are based on bibliography research.
Authors' response:
- With some additional experiments we addressed potential specific effects of miRNA 425-5p, as this miRNA has several target genes relevant for GBM biology, among them GFAP, the marker we used to describe differentiated tumor cells in contrast to GBM stem-like cells. We could show that a mimic-miR425-5p transfected into differentiated GBM cells is able to suppress GFAP expression.
- We also demonstrate the individual miRNA expression data in the GSC lines used (Figure 3 and Supplemental Figure 1).
- Globally, we demonstrated for the 10 most regulated miRNAs a KEGG analysis and a gene regulation network by bioinformatic analysis. For this, we received help from the Center for Omics Science at San Raffaele, Milan, and added one author (Raffaella Pini)
Specific comments:
- In the introduction (line 39) the authors describe/define GSCs as cells that express a unique marker (CD133). This is a poor definition of GSCs; a better and wider contextualization of these cells is needed, in terms of a combination of different stem cells markers, heterogeneity within the stem cell compartment and/or the natural biology for the transition from Proneural to Mesenchymal upon GBM progression.
Authors' response: We have addressed this comment by adding more potential stem cell markers such as Sox2, Nestin, and CD44 by qPCR and Sox2 by Western Blot. We were able to describe the GSCs by a significant dowregulation of Sox2, both, on the mRNA and protein level. To crosscheck for differentiation, we also detected protein expression of GFAP which was more abundant in differentiated cells. Our additional description of the GSC phenotype are now shown in a new Figure 1.
- Table 1 itself and also the table 1 legend are allocated in the Materials and Methods section, the should move to Results.
Authors' response: This is now fixed.
- Table 2 legend also should be moved to the bottom of Table 2.
Authors' response: This is now fixed.

Reviewer 2 Report
This research aimed to detect the changes of the miRNA expression profiles in patient-derived GSCs altered upon differentiation into glial fiber acid protein (GFAP) expressing, astrocytic tumor cells using a polymerase chain reaction (PCR) array. However, the data need to be further validation in vitro and in vivo, so at this stage, the conclusion is premature. Also, there were many miRNAs altered before and after differentiation, only chose four miRNAs that were reported in GBM make this research further lose its novelty.
Author Response
Dear Reviewer,
thanks for your productive comments on our manuscript. In our revised version, we have addressed your comments as follows:
This research aimed to detect the changes of the miRNA expression profiles in patient-derived GSCs altered upon differentiation into glial fiber acid protein (GFAP) expressing, astrocytic tumor cells using a polymerase chain reaction (PCR) array. However, the data need to be further validation in vitro and in vivo, so at this stage, the conclusion is premature. Also, there were many miRNAs altered before and after differentiation, only chose four miRNAs that were reported in GBM make this research further lose its novelty.
Authors' response:
We thank the reviewer for this productive and encouraging comment.
- We provide further validation of the stem cell phenotype by adding more stem cell markers such as Sox2 at the protein level which is now shown in Figure 1. We also tried qPCR on other suggested marker such as CD44 and Nestin, however the results were not conclusive (please see attached file "qPCR_Stem_Cell_Markers")
- We have now analyzed the individual changes in miRNA expression levels in all three GSC lines.
- We provide a table with the 10 most regulated miRNAs.
- On these 10 miRNA we provide a bioinformatic analysis of their gene regulation network (by KEGG and Chord analyses).
- We have now added data on the functional characterization of miRNA 425-5p, a candidate miRNA for regulation of the GFAP gene.

Reviewer 3 Report
Glioblastoma (GBM) is a common type of adult brain tumor. In this study, Evers et al., studies the miRNAs in CD133+ glioblastoma-stem-like cells (GSCs), the CD133-high, self-renewing cells that are capable of multi-lineage differentiation, by comparing with the differentiated counterparts. They used miRNA PCR array to study the changes of miRNA profiles between pooled samples from three patient-derived GSCs and their differentiated cells, and identified 22 miRNAs with higher expression in GSCs and 9 miRNAs with higher expression in differentiated cells. They further implement literature search and target prediction analyses to generate a short list of miRNAs that might be worth further study. Overall, this study is well-designed and performed. The data are presented in a scientific fashion. The list of candidate miRNAs found in this study will be helpful for relevant follow-up research. I recommend the acceptance of the study once the authors address the below comments / suggestions.
1) In addition to CD133 and GFAP, I suggest the authors to add another 1-2 markers for the validation of GSCs and differentiated cells, such as CD44 and SOX2.
2) Since the miRNA array was done in pooled samples. It is worth validating several differentially expressed miRNAs in each sample using RT-qPCR, so that to evaluate the variation among three samples.
3) For the heatmaps in Figure 2, color scale indicates that green is for negative values which is for down-regulated miRNAs. It’s better that the authors state the comparison more clearly in the figure and the legend, that green is for down-regulated ones in differentiated cells which means the up-regulated ones in GSCs.
Author Response
Dear Reviewer,
thanks for your productive comments on our manuscript. In our revised version, we have addressed the general and specific comments as follows:
1) In addition to CD133 and GFAP, I suggest the authors to add another 1-2 markers for the validation of GSCs and differentiated cells, such as CD44 and SOX2.
Authors' response: thanks for this advice. We have done qPCR analyses using CD44, SOX2, and nestin as GSC markers. The results of these qPCR experiments are attached ("qPCR_Stem_Cell_Marker.pdf"). In addition, we used Sox2 protein as an additional GSC marker. These results are now shown in a new Figure 1. Downregulation of Sox2 correlates nicely with the induction of GFAP expression.
2) Since the miRNA array was done in pooled samples. It is worth validating several differentially expressed miRNAs in each sample using RT-qPCR, so that to evaluate the variation among three samples.
Authors' response: As suggested, we have now performed all miRNA analyses in each GSC line to validate the miRNA array data. For miR-425-5, the results are shown in a new Figure 3, for all other miRNAs in a new Supplemental Figure 1.
3) For the heatmaps in Figure 2, color scale indicates that green is for negative values which is for down-regulated miRNAs. It’s better that the authors state the comparison more clearly in the figure and the legend, that green is for down-regulated ones in differentiated cells which means the up-regulated ones in GSCs.
Authors' response: we apologize for this unclarity. We have now added more information on the color coding.

Round 2
Reviewer 1 Report
The authors have adressed each of the comments reaised in the revisions and have improved the
Author Response
Thanks for this positive response, we appreciate your help.
Reviewer 2 Report
All question were answered.
Author Response
Thanks.
Reviewer 3 Report
In the rebuttal letter, the authors stated that ‘We have done qPCR analyses using CD44, SOX2, and nestin as GSC markers. The results of these qPCR experiments are attached ("qPCR_Stem_Cell_Marker.pdf").’ However, I didn’t see this file and I also wonder why they didn’t include these results in the manuscript? Or am I missing something?
Author Response
We have attached these data to our reply, and apologize that you have missed them. To show the data as part of the manuscript, we have now decided to include these data in Supplementary Figure 1 (the original Supplementary Figure 1 is now Supplementary Figure 2). The decrease in mRNA levels of stem cell markers nestin and Sox2 is not seen in all three GSCs, and CD44 is induced in differentiated GSCs.